

# A literature review of research on question generation in education

Xiaohui Dong, Xinyu Zhang, Zhengluo Li, Quanxin Hou, Jixiang Xue and Xiaoyi Li

College of Computer Science & Engineering, Northwest Normal University, Lanzhou, Gansu, China

## ABSTRACT

As a key natural language processing (NLP) task, question generation (QG) is crucial for boosting educational quality and fostering personalized learning. This article offers an in-depth review of the research advancements and future directions in QG in education (QGEd). We start by tracing the evolution of QG and QGEd. Next, we explore the current state of QGEd research through three dimensions: its three core objectives, commonly used datasets, and question quality evaluation methods. This article also underscores its unique contributions to QGEd, including a systematic analysis of the research landscape and an identification of pivotal challenges and opportunities. Lastly, we highlight future research directions, emphasizing the need for deeper exploration in QGEd regarding multimodal data processing, controllability of fine-grained cognitive and difficulty levels, specialized educational dataset construction, automatic evaluation technology development, and system architecture design. Overall, this review aims to provide a comprehensive overview of the field, offering valuable insights for researchers and practitioners in educational technology.

## INTRODUCTION

Formulating questions is an indispensable part of the educational process. Well-crafted questions can not only inspire students to grasp the content of teaching but also check students' understanding of knowledge, stimulate their interest in learning, and promote the development of their thinking. In traditional educational models, the design of questions is usually manually completed by subject teachers or educational experts. This method requires a significant amount of time and cognitive effort, and the quality of the questions is largely limited by the professional level and scope of attention of the designers. With the rapid growth of knowledge and information, this traditional manual questioning approach has become difficult to adapt to the needs of large-scale online learning and ubiquitous learning in the current digital age. Automatic question generation (AQG) technology in education can assist in developing teaching resources, promote personalized teaching, enhance educational quality, and support teachers' professional development. In summary, AQG has tremendous potential to empower education, and AQG in education holds significant research and application value.

Corresponding author
Xiaohui Dong, dxh@nwnu.edu.cn

**Table 1 The instance from the XQuAD (*Artetxe, Ruder & Yogatama, 2020*) dataset suitable for answer-aware question generation.**

**Input Paragraph:** The heat required for boiling the water and supplying the steam can be derived from various sources, most commonly from [**burning combustible materials**]₁ with an appropriate supply of air in a closed space (called variously [**combustion chamber**]₂, firebox). In some cases, the heat source is a nuclear reactor, geothermal energy, [**solar**]₃ energy or waste heat from an internal combustion engine or industrial process. In the case of model or toy steam engines, the heat source can be an [**electric**]₄ heating element.

**Answer Span:** burning combustible materials, combustion chamber, solar, electric

**Question1:** What is the usual source of heat for boiling water in the steam engine?
**Question2:** Aside from firebox, what is another name for the space in which combustible material is burned in the engine?
**Question3:** Along with nuclear, geothermal and internal combustion engine waste heat, what sort of energy might supply the heat for a steam engine?
**Question4:** What type of heating element is often used in toy steam engines?

AQG abbreviated as question generation (QG), has its research history that dates back to the 1970s (*Wolfe, 1976*). It is an important and challenging task in cognitive intelligence, aimed at enabling computers to generate content-relevant and fluently expressed questions from various inputs, such as natural language texts (*Zhang et al., 2022*), knowledge bases (*Reddy et al., 2017*; *Guo et al., 2023*; *Bi et al., 2024*), images (*Mulla & Gharpure, 2023*), and videos (*Priya et al., 2022*). This reflects the process of a machine moving from understanding knowledge, discovering knowledge, to applying knowledge. QG can generally be formally defined as: given a natural language text $D = \{x_1,\ldots,x_n\}$, the goal is to generate natural language questions $Q = \{y_1,\ldots,y_{|y|}\}$, that is:

$$\begin{aligned}\overline{Q} &= \arg\max P(Q|D) \\ &= \arg\max_y \sum_{i=1}^{|y|} \log P(y_i|D, y_{<i})\end{aligned} \tag{1}$$

$x_i$ denotes a word within the input text, $n$ signifies the length of the input text, $y_i$ refers to a word within the generated question, and $|y|$ indicates the sentence length of the generated question (*Zhang et al., 2022*).

There are multiple ways to classify QG tasks. Typically, these tasks are categorized based on whether the focus is on the answer to the question, that is, whether the input contains an answer, into answer-aware (answer-sensitive) and answer-unaware QG. In answer-aware QG, questions are formulated towards a given answer within the context, as shown in Table 1, where the input paragraph represents the provided context, and the answer span indicates the position of the target answer within it. These answers also serve as inputs for answer-aware QG. Question1 to question4 are the corresponding questions generated by the machine based on the provided context and the target answer. In contrast, answer-unaware QG is not constrained by a target answer and only requires the formulation of questions based on the given context information (document, paragraph, sentence, or keywords), as shown in Table 2, where the input document is the provided context, and question1 to question4 are the questions generated by the machine. The answers to these questions may either be present in the original text or simply related to its content.

**Table 2 The instance from the LearningQ (*Chen et al., 2018*) dataset suitable for answer-unaware question generation.**

**Input Document:** … Why is this region named this way? What is it in the middle of or near to? It is the proximity of these countries to the West (to Europe) that led this area to be termed "the near east." Ancient Near Eastern Art has long been part of the history of Western art, but history didn't have to be written this way. It is largely because of the West's interests in the Biblical "Holy Land" that ancient Near Eastern materials have been regarded as part of the Western canon of the history of art …

**Question1:** So, Mesopotamia is a name that could be used interchangeably with the Middle East or the Near East?
**Question2:** How does it compare to a city like New York city?
**Question3:** What were the conventions of art in the Ancient Middle East?
**Question4:** What about the lost gardens of Babylon?

With the gradual maturation of general-purpose large language models (LLMs), some studies have begun to explore prompt engineering as a less costly and more efficient approach to guide these models in generating the desired questions. For instance, *Lee et al. (2024)* employed few-shot prompting to direct the model to generate questions of varying cognitive levels within a 2D matrix framework that encompasses multiple question types and formats. *Tonga, Clement & Oudeyer (2024)* designed error-based prompts and general prompts based on the principles of chain-of-thought prompting, enabling the "teacher LLM" to generate heuristic questions in response to the reactions of the "student LLM".

## Rational

Regarding QG in education (QGEd), there are also some relevant reviews. We found that some reviews did not explicitly propose the characteristics and objectives of QGEd, or they were not sufficiently focused on the educational field. For instance, *Le, Kojiri & Pinkwart (2014)* reviewed the research progress of QGEd, categorizing it into three types from the perspective of application objectives: knowledge/skill acquisition, knowledge evaluation, and tutoring dialogues. *Kurdi et al. (2020)* also provided a review of this field, but there was insufficient discussion on the guidance and integration of educational theories in generating questions. In 2024, *Al Faraby, Adiwijaya & Romadhony (2024)* conducted a review of QGEd from 2016 to the beginning of 2022, focusing on analyzing the technical components of QG, such as context, answers, questions, and datasets. Although this study concentrated on reviewing methods based on neural network models and involved a relatively small number of literatures, it clearly proposed that naturalness, diversity, controllability, practicality (educational value), and personalization are the core objectives of research in this field, providing a clear direction for QGEd research:

(1) Naturalness means that the generated questions should not only conform to the expression habits and grammatical rules of the target language but also ensure that the questions can be answered from the context or at least can be resolved using background knowledge related to the context. This goal is a universal requirement for all QG tasks, not limited to QGEd.

(2) Diversity refers to generating a variety of different types of questions for the same context to avoid producing overly general and similar questions, in order to meet the needs of different teaching scenarios.

(3) Controllability mainly points to the cognitive level and difficulty level of the generated questions, that is, guiding the model to generate questions of different cognitive levels or difficulty levels through corresponding mechanisms.

(4) Educational value emphasizes that the generated questions should focus on teaching objectives and content, avoiding the generation of some popularized and trivial questions.

(5) Personalization aims to meet the individual differences of learners and specific educational needs.

Among these objectives, naturalness has been well addressed in previous research. For instance, in studies related to heuristic rule-based methods, language features are often utilized, such as named entity recognition (*Sang & Meulder, 2003*), positional tagging (*Lindberg et al., 2013*), or by using placeholders (*Jia et al., 2021*; *Kumar, Banchs & D'Haro, 2015*; *Olney, Pavlik & Maass, 2017*) to replace keywords or sentences in the context; in neural question generation (NQG) methods, many works have improved the quality of questions by refining the encoding and decoding processes (*Wang et al., 2019*; *Song et al., 2018*; *Kim et al., 2019*). Furthermore, some studies have adopted strategies such as question ranking (*Zheng et al., 2018*), reinforcement learning (*Fan et al., 2018*), and multi-task learning (*Wang, Yuan & Trischler, 2017*) to further enhance the naturalness of the generated questions. And personalization is a comprehensive embodiment of other objectives. In summary, this study believes that diversification, controllability, and focus on teaching content should be the core objectives of QGEd.

In terms of literature breadth, this review covers more QGEd research articles than previous reviews, totaling 48. Regarding the review's scope, it is more comprehensive and systematic than previous work, covering the development of QG core technologies, the current state of research on QGEd's three core objectives, datasets, question evaluation schemes, and future research directions. In terms of perspective, this review starts from educational theories and practical needs, deeply discussing the achievements and shortcomings of current technical research, thereby providing better guidance for its future development. In summary, this review introduces existing QGEd technologies developed by other researchers, defines their core algorithms and distinctive features, and helps researchers gain a thorough understanding of the current state of QGEd development, thereby inspiring them to take the next steps in their work.

### Intended audiences

This review is intended for two main groups of audiences with varying levels of expertise in computer. The first group consists of non-technical individuals, such as teachers and educational researchers, who may not have extensive experience with computer techniques. The second group includes technical audiences, such as education technology researchers and developers in the artificial intelligence industry, who may have more familiarity with these techniques. The first objective is to motivate teachers and educational researchers who have relied heavily on conventional question design and lack an understanding of computer vision algorithms. We aim to encourage them to engage

with question design used in practical teaching and teaching research. The second objective is to assist and inform education technology researchers and developers about the current state of QGEd. This allows them to take inspiration from existing work and make modifications, thereby preventing them from having to reinvent the wheel.

The rest of this article is organized as follows. The methodology used to conduct this survey is discussed in 'Survey Methodology'. The progress of QG from the perspective of technological evolution and the evolution trends of QGEd are introduced in 'Development of Technology'. The research progress of QGEd in achieving the three core objectives is discussed in 'Three Core Objectives'. An overview of typical datasets commonly used for QG is presented from the perspective of QGEd in 'Datasets', followed by a review of commonly used evaluation schemes for QGEd in 'Evaluation Schemes'. Finally, discussions and future work directions are presented in 'Discussions', and the overall conclusions are summarized in 'Conclusion'.

## CONTRIBUTIONS

This article makes several key contributions to the field of QGEd:

(1) We provide a systematic analysis of the research landscape of QGEd, tracing its evolution from heuristic rule-based methods to neural network technologies.
(2) We identify pivotal challenges and opportunities in QGEd, including the need for multimodal data processing, controllability of fine-grained cognitive and difficulty levels, and specialized educational dataset construction.
(3) We propose future research directions to address these challenges, emphasizing the development of automatic evaluation technologies and system architecture design.
(4) We offer a comprehensive overview of the field, synthesizing insights across its evolution, core objectives, datasets, and evaluation methods, which provides a clear understanding of the current state and progress for researchers and practitioners in educational technology.

## SURVEY METHODOLOGY

This study was conducted following the guidelines of the 2018 PRISMA framework for scoping reviews (*Tricco et al., 2018*), which provides a set of rigorous and transparent methods to ensure trustworthy results. This review aims to collect recently published studies in order to analyze the development of QGEd research over the past decade (2015–2025). Due to the special timing of the manuscript, the studies published in 2025 have not been fully collected in this review.

### Information source and search process

The five scientific databases were considered during the initial phase (see Fig. 1): Web of Science, IEEE Explore Digital Library, Springer Link, DBLP Computer Science Bibliography and ACM Digital Library. Each database includes relevant studies about QGEd. The search strategy included limiting the search results to studies that were published between 2015 and 2025 and that were written in English. The keywords chosen

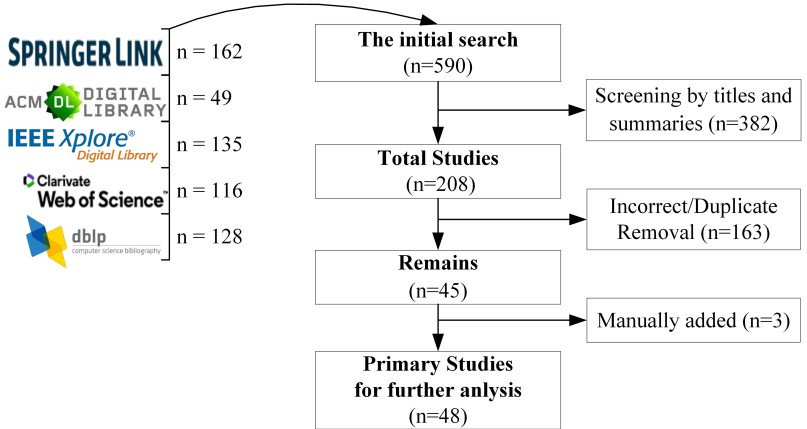

**Figure 1 Summary of scoping review databases.**

to identify the relevant records were "question generation," "question generator," "question generation for education," "difficulty-controllable generation," "question-type driven question generation," "diverse question generation," "Automatic education QG." It should be noted that this review primarily focuses on studies related to English QG, as the majority of the literature retrieved from citation databases centers on English. However, this does not imply that research on QG in other languages is excluded, which may also provide valuable insights and directions for future exploration.

## Articles selection

The initial search on those five databases returned 590 articles. The titles and summaries of these articles were screened to exclude irrelevant works. As a result of this first screening, 208 articles were selected and considered for the next screening phase, which included a full review of the articles to check whether the eligibility criteria were met. Finally, 48 unique and fully accessible studies were identified as primary sources for further analysis in this review. Details about the number of records retrieved by each scientific database and the process of the data extraction and monitoring are shown in Fig. 1.

## Data items

Each selected articles were indexed in a local database and, for each study, the following characteristics were included: title, year, core technology (heuristic rule, neural network or the combination of these two), core objective, database, evaluation metrics and evaluation/ assessment strategies. Such characteristics were deemed relevant to reach the aim of the present work.

## Synthesis of results

The authors of this article, based on the selected articles, provides a detailed and comprehensive review of the technological development trajectory, the current status of research on the three core objectives, commonly used datasets in research, and evaluation schemes of QGEd in the rest of sections. The basic information of the selected articles is

**Table 3 The basic information of the surveyed papers about QGEd from January 2015 to May 2024.**

| Articles | Core technology | Diversification | Controllability | Focus on teaching content | Subject | Assessment scheme |
|---|---|---|---|---|---|---|
| *Kumar, Banchs & D'Haro (2015)* | Heuristic rule & NN | √ | | | Biology | Manual |
| *Olney, Pavlik & Maass (2017)* | Heuristic rule | √ | √ | √ | Medicine | Manual |
| *Du, Shao & Cardie (2017)* | NN | | √ | | English | Manu+Auto |
| *Yang et al. (2017)* | NN | √ | √ | | Multidisciplinary | Automatic |
| *Du & Cardie (2017)* | NN | √ | √ | | English | Automatic |
| *Song et al. (2018)* | NN | | √ | | English | Automatic |
| *Flor & Riordan (2018)* | Heuristic rule | | √ | | English | Manual |
| *Park, Cho & Lee (2018)* | Heuristic rule | √ | | | English | Manual |
| *Gao et al. (2019)* | NN | | √ | | English | Manu+Auto |
| *Du & Cardie (2018)* | NN | | √ | √ | English | Manu+Auto |
| *Sun et al. (2018)* | NN | √ | √ | √ | English | Automatic |
| *Wang et al. (2019)* | NN | | √ | | English | Manu+Auto |
| *Chan & Fan (2019)* | NN | | √ | √ | English | Automatic |
| *Wang et al. (2020a)* | NN | √ | √ | √ | English | Automatic |
| *Liu et al. (2020)* | NN | √ | | | English | Manu+Auto |
| *Steuer, Filighera & Rensing (2020)* | NN | | | √ | English | Manu+Auto |
| *Jia et al. (2021)* | NN | | √ | √ | English | Manu+Auto |
| *Srivastava & Goodman (2021)* | NN | | √ | | Language learning | Manu+Auto |
| *Cheng et al. (2021)* | NN | | √ | √ | Multidisciplinary | Manu+Auto |
| *Murakhovs'ka et al. (2021)* | NN | √ | √ | | Multidisciplinary | Manu+Auto |
| *Cao & Wang (2021)* | NN | | √ | | Multidisciplinary | Manu+Auto |
| *Qu, Jia & Wu (2021)* | NN | | √ | | English | Manu+Auto |
| *Stasaski et al. (2021)* | Heuristic Rule & NN | | √ | | Multidisciplinary | Manu+Auto |
| *Cui et al. (2021)* | NN | √ | | √ | Multidisciplinary | Manu+Auto |
| *Steuer et al. (2021)* | Heuristic Rule & NN | | | √ | Multidisciplinary | Manual |
| *Zou et al. (2022)* | Heuristic Rule & NN | √ | √ | | English | Manual |
| *Ch & Saha (2023)* | NN | √ | √ | √ | Multidisciplinary | Manual |
| *Zeng et al. (2022)* | NN | | √ | | Multidisciplinary | Automatic |
| *Liu & Zhang (2022)* | NN | √ | | √ | Multidisciplinary | Manu+Auto |
| *Yao et al. (2022)* | Heuristic Rule & NN | | √ | √ | Fairy Tale | Manu+Auto |
| *Guo et al. (2023)* | NN | √ | | | Multidisciplinary | Manu+Auto |
| *Gou et al. (2023)* | NN | √ | | | Multidisciplinary | Manu+Auto |
| *Chomphooyod et al. (2023)* | Heuristic Rule & NN | √ | √ | | English | Manu+Auto |
| *Matsumori et al. (2023)* | NN | √ | √ | √ | English | Manu+Auto |
| *Prokudin, Sychev & Denisov (2023)* | Heuristic Rule & NN | | √ | | Programing | Manual |
| *Bi et al. (2024)* | NN | √ | √ | | Multidisciplinary | Manu+Auto |
| *Babakhani et al. (2024)* | NN | √ | | | Multidisciplinary | Manu+Auto |
| *Bitew et al. (2024)* | NN | √ | √ | | Multidisciplinary | Manual |

| Articles | Core technology | Diversification | Controllability | Focus on teaching content | Subject | Assessment scheme |
|---|---|---|---|---|---|---|
| *Duong-Trung, Wang & Kravcik (2024)* | NN | | √ | √ | Introduction to machine learning & neural networks and deep learning | Manu+Auto |
| *Zhang, Xie & Qiu (2024)* | NN | | √ | √ | English | Manu+Auto |
| *Moon et al. (2024)* | NN | | √ | | English | Manu+Auto |
| *Teruyoshi, Tomikawa & Masaki (2024)* | NN | | √ | | English | Automatic |
| *Tomikawa & Masaki (2024)* | NN | | √ | | English | Automatic |
| *Tomikawa, Suzuki & Uto (2024)* | NN | | √ | | English | Manu+Auto |
| *Tonga, Clement & Oudeyer (2024)* | Prompt Engineering | √ | √ | √ | Math | Automatic |
| *Lee et al. (2024)* | Prompt Engineering | √ | √ | √ | English | Manu+Auto |
| *Shoaib et al. (2025)* | NN | | √ | √ | Computer | Manu+Auto |
| *Maity, Deroy & Sarkar (2025)* | Prompt Engineering | √ | √ | √ | Educational content | Manu+Auto |

**Note:**
NN refers to Neural Network. Manu refers to Manual. Auto refers to Automatic.

also organized in the Table 3 for readers to quickly and conveniently locate the reference literature that meets their needs.

# DEVELOPMENT OF TECHNOLOGY

The exploration of general-purpose AQG technologies has established a robust technical foundation for QGEd. This section is dedicated to categorizing these general QG techniques and summarizing their evolutionary trends, with the aim of offering future researchers a technical roadmap for QGEd studies.

Regardless of the type of QG task, from the perspective of technological evolution, they can generally be divided into three stages: heuristic rule-based methods, neural question generation (NQG) and prompt engineering.

## Heuristic rule-based methods

Heuristic rule-based methods were the mainstream approach in early question generation. These methods typically rely on artificially designed transformation rules to convert given contexts into corresponding questions. They can generally be divided into three categories: template-based methods (*Mostow & Chen, 2009*; *Zheng et al., 2018*; *Fan et al., 2018*; *Liu et al., 2019*; *Fabbri et al., 2020*), grammar-based methods (*Wolfe, 1976*; *Kunichika et al., 2004*; *Mitkov & Ha, 2003*; *Heilman & Smith, 2010*; *Ali, Chali & Hasan, 2010*), and semantics-based methods (*Huang & He, 2016*; *Yao & Zhang, 2010*; *Flor & Riordan, 2018*; *Dhole & Manning, 2020*). Generally, these methods include two steps: content selection and question construction. First, the topic to be inquired about is selected through semantic or syntactic parsing methods, and then templates or

transformation rules are used to convert the context of the selected topic into a natural language question.

### Template-based methods

Template-based methods are commonly used for QG in closed-domain specific applications. For instance, *Zheng et al. (2018)* proposed a template technique for constructing questions from Chinese text. *Wolfe (1976)* designed a pattern matching method for use in educational systems. *Fabbri et al. (2020)* employed a template method to perform sentence extraction and question generation in an unsupervised manner.

### Grammar-based methods

Grammar-based methods first determine the syntactic structure of the given text, and then apply syntactic transformation rules and the part-of-speech of interrogative sentences to generate questions. *Mitkov & Ha (2003)* were among the first to attempt generating multiple-choice questions using transformation rules. *Heilman & Smith (2010)* employed an overgeneration and ranking strategy for generating factual questions. *Ali, Chali & Hasan (2010)* used syntactic analysis, part-of-speech tagging, and named entity recognizers to generate questions.

### Semantics-based methods

Semantics-based methods generate questions by performing semantic analysis on the text. *Huang & He (2016)* introduced lexical functional grammar (LFG) as a linguistic framework for QG. *Yao & Zhang (2010)* proposed a semantics-based method for QG based on minimal recursion semantic representation. *Lindberg et al. (2013)* used a template method incorporating semantic roles to generate questions that support online learning. *Flor & Riordan (2018)* also generated factual questions based on semantic roles. *Dhole & Manning (2020)* developed Syn-QG, which utilizes dependency relations, shallow semantic parsing, and custom rules to generate questions.

## Neural question generation

In 2017, *Du & Cardie (2017)* first applied the sequence-to-sequence (Seq2Seq) encoder-decoder model to question generation. Since then, neural question generation (NQG) has become the mainstream approach in QG research, leading to a surge of high-quality studies and achieving more desirable results. Most of the work in NQG utilizes recurrent neural networks (RNN) or transformer networks with attention mechanisms to automatically learn patterns and relationships in the input text, capturing complex dependencies and generating questions that may not be explicitly defined in rules, adapting to different domains when there is sufficient training data available.

### RNN-based methods

RNN-based models consist of an encoder and a decoder, both of which are RNNs, such as long short-term memory (LSTM). These models typically include a word embedding layer, an encoder layer, and a decoder layer, equipped with attention mechanisms and copying mechanisms. For instance, *Chali & Baghaee (2018)* incorporated the coverage mechanism into the RNN-based framework. *Tang et al. (2018)* introduced copying and

post-processing mechanisms into the RNN-based framework to address the issue of rare or unknown words. *Reddy et al. (2017)* utilized the RNN-based framework to generate questions from a given set of keywords.

### Transformer-based methods

The main difference between the Transformer and RNN-based models is that the transformer's encoder and decoder are entirely implemented by attention mechanisms. In answer-unaware QG, *Kumar et al. (2019)* proposed a cross-lingual model based on the Transformer encoder-decoder architecture to improve QG in the target language. *Scialom, Piwowarski & Staiano (2019)* suggested incorporating copying mechanisms, placeholder strategies, and Embeddings from Language Models (ELMo) into the Transformer. *Pan et al. (2020)* introduced contextual information and control signals into the Transformer-based Seq2Seq model to generate diversified Chinese questions from keywords. Using an answer as input for a QG system can generate more specific and context-related questions; such systems are known as answer-aware QG. In answer-aware QG, *Chan & Fan (2019)* employed the Transformer-based BERT model to address the QG task with answer position embedding. *Chai & Wan (2020)* proposed a Transformer-based approach that utilizes answer span information to generate questions in an autoregressive manner.

### Pre-trained model based methods

As large-scale pre-trained language models have achieved remarkable results in various natural language processing tasks, fine-tuning pre-trained language models to adapt to downstream tasks has become a research hotspot. For instance, *Dong et al. (2019)* proposed a model called UniLM that is pre-trained using three language modeling tasks. *Wang et al. (2020b)* introduced a simple and effective knowledge distillation model called MiniLM for compressing large pre-trained models. *Bao et al. (2020)* utilized masking strategies to learn relationships between terms and pseudo-masking to learn relationships between segment contents. *Qi et al. (2020)* designed a pre-trained model aimed at simultaneously predicting n-gram terms, applying an n-stream attention mechanism. *Xiao et al. (2020)* proposed a multi-stage pre-training and fine-tuning framework called ERNIE-GEN. *Wu et al. (2021)* used a pre-trained BERT model for question type classification and integrated a penalty mechanism for QG. *Yulan et al. (2022)* addressed the "exposure bias" and "masking heterogeneity" issues in the decoding phase of pre-trained models by employing methods based on stochastic noise resistance and transfer learning, enhancing the adaptability of the pre-trained model UniLM in QG tasks.

### Other neural-network based methods

In addition to the main QG methods mentioned above, some researchers have also employed machine learning methods such as graph models, generative adversarial networks, and variational autoencoders (VAEs). For instance, *Liu et al. (2019)* used graph convolutional networks to represent the syntactic dependency tree of paragraphs. *Ma et al. (2020)* constructed an answer-centric entity graph from documents and utilized graph convolutional networks (GCN) to aggregate semantic information at different

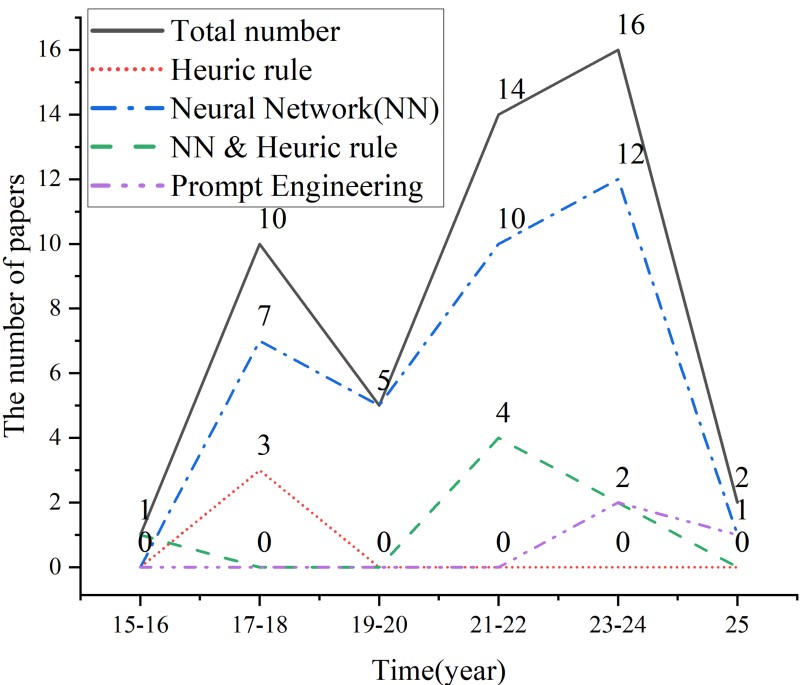

**Figure 2  The core technology evolution trends in QGEd.**

granularities. *Rao & Daumé (2019)* employed generative adversarial networks to refine QG. *Zhou et al. (2023)* introduced iterative graph neural networks (GNN) in the decoding stage to address the issue of losing rich sequential and semantic structural information from the paragraph during the decoding process.

### *Prompt engineering*

The crux of prompt engineering lies in the meticulous design of input prompts to elicit outputs from LLMs that meet specific requirements. This approach obviates the need for updating the model's weights or parameters. Instead, it adeptly constructs prompts to steer the model towards generating questions in the desired direction (*Lee et al., 2024*; *Tonga, Clement & Oudeyer, 2024*; *Maity, Deroy & Sarkar, 2025*).

### The core technology evolution trends in QGEd

To obtain high-quality data for the research, this study utilizes widely recognized scientific literature databases, including Web of Science, ACM, dblp, IEEE, and Springer. The search period is limited to January 2015 to February 2025, and a total of 48 QGEd research articles were collected. Based on the publication years of the literature, QGEd experienced a research peak between 2017 and 2018, as shown in Fig. 2. During this period, the number of studies using deep neural network models peaked at 7; in contrast, the number of studies based on heuristic rules rapidly decreased. After 2020, heuristic rules were primarily used to address the limitations of neural network models in controlling generation factors. Moreover, in recent years, prompt engineering has emerged as a novel and highly efficient method for guiding LLMs to automatically generate questions.

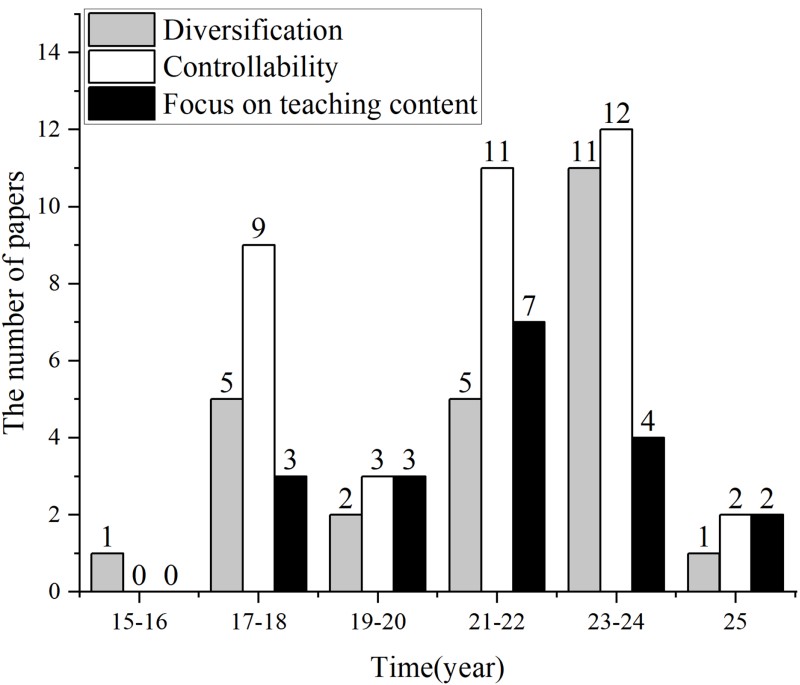

**Figure 3 Distribution of studies on the three core objectives.**

# THREE CORE OBJECTIVES

In this section, we will delve into the current state of development of QGEd with respect to its three core objectives: diversification, controllability, and focus on teaching content.

QGEd, as a subfield of QG specifically tailored for educational applications, has benefited from the technological advancements in QG, which have laid the groundwork for its implementation. However, the unique demands of the educational scenario have set higher standards for QGEd. As deep neural network models have become the main approach in this field, achieving the special goals of QGEd has gradually become possible. As shown in Fig. 3, the quality and density of research related to diversification, controllability, and focus on teaching content have significantly improved after 2017, with controllability receiving more attention from researchers than the other objectives.

## Diversification

In the field of education, the diversification of question design is a widely recognized need. Diversified questions not only promote learners' thinking at different cognitive levels and deepen their understanding of the learning content, but also stimulate learning interest and creativity, and cultivate a multi-perspective approach to problem-solving. In the field of QGEd, question diversification is mainly reflected in two aspects: expression diversification and question type diversification (*Zhang & Zhu, 2021*). Expression diversification refers to conveying the same semantics through different expressions, and these questions, although stated in various ways, all point to the same answer. This kind of diversification helps to cultivate and exercise students' transferability and semantic

discernment abilities. Question type diversification, on the other hand, involves the use of different types of questions, such as fill-in-the-blank, multiple-choice, short answer, and true/false questions, to meet different teaching needs and learners' learning styles.

### Expression diversification

VAEs are widely adopted methods, representing a type of generative model for latent variables. Typically, related research utilizes an encoder to map observed variables (*i.e.*, questions along with their corresponding answers and context) to latent variables, and then employs a decoder to map the latent variables back to observed variables. The training of this model is usually accomplished by maximizing the variational lower bound and some auxiliary objectives. For instance, *Wang et al. (2020a)* adopt a multimodal prior distribution strategy, selecting diversified background information related to the answer through a conditional variational autoencoder (CVAE), modeling it as continuous latent variables. They also explicitly model the expression types of questions, guiding the model to generate questions with diverse expressions through these auxiliary pieces of information. *Shinoda & Aizawa (2020)* proposed a variational QA pair generation model, introducing two independent latent random variables to model the one-to-many problem in QG. *Lee et al. (2020)* proposed a hierarchical conditional variational autoencoder (HCVAE), designing two independent latent variable spaces for questions and answers conditioned on input text.

Some studies simplify the QG task to generating diversified questions based on preset keywords to facilitate the generation process. These studies typically use keyword groups as input for the generation model, training it to complete the one-to-many QG task. For example, *Pan et al. (2020)* combine multi-head self-attention mechanisms with contextual information (questions that have already been generated, rather than the original text containing the keywords) and introduce control signals to increase question diversity while ensuring high relevance to the given keywords. *Liu et al. (2020)* adopt a similar approach, proposing an answer-clue-style aware question generation (ACS-QG) method. This method automatically constructs a dataset of (answer, clue, question word type) triplet structures from the SQuAD dataset to train the QG model. It further demonstrates the key role of different questioning clues (*i.e.*, answer background information) in the diversity of question expression while enhancing the controllability of question expression methods.

Other studies have demonstrated that introducing external information or combining templates with neural network models can also benefit the generation of diversified questions. For instance, *Liu et al. (2017)* combined template-based methods with Seq2Seq learning to generate fluent and diverse questions. *Guo et al. (2023)* introduced a dual-model framework interwoven with two selection strategies, which extracts more diverse expressions from external natural questions and integrates them into the generation model to enhance question diversity. *Gou et al. (2023)* further considered the impact of external knowledge on question diversity, proposing a retrieval-enhanced type transformation framework (RAST) that utilizes type-diverse templates for question generation. *Zou et al. (2022)* proposed an unsupervised true/false QG method (TF-QG)

that enables the QG model to create diverse judgment questions using heuristic templates generated by various NLP techniques.

### Question type diversification

In the research on diversified question type generation, early heuristic rule-based question generation techniques deeply studied the grammar and semantics of sentences, enabling the generation of simple factual fill-in-the-blank questions and cloze tests from given texts (*Kumar, Banchs & D'Haro, 2015*; *Olney, Pavlik & Maass, 2017*). However, due to the difficulty of these strategies in overcoming the challenges of complex natural language processing (NLP), researchers began to seek alternative methods. With the significant advantages of deep neural networks in handling complex issues, more and more researchers are adopting neural network models to enhance the diversification of question types in QG. For example, *Park, Cho & Lee (2018)* used text embedding models to project documents, sentences, and word vectors into the same semantic space, thereby identifying sentences and words with educational value. After ranking the similarity of these elements, they were transformed into interrogative sentences and corresponding blanks for the interrogative words. This method can generate multiple-choice fill-in-the-blank questions without relying on any prior knowledge or additional datasets. *Chomphooyod et al. (2023)* used a text generation model trained with the Text-to-Text Transfer Transformer (T5) architecture to control the content and grammatical themes through keywords and part-of-speech (POS) templates. They also used rule-based algorithms to generate correct answers and distractors for 10 grammatical themes, thereby generating grammatical multiple-choice questions. The CLOZER algorithm introduced by *Matsumori et al. (2023)* utilizes a masked language model (MLM) and a new metric, gap score, to automatically generate educationally valuable open cloze questions (OCQ).

## Controllability

Controllable QGEd primarily refers to the controllability of question difficulty and cognitive level. Difficulty and cognitive level are two interrelated but independent concepts. In the teaching process, it is necessary to design questions of different cognitive levels for the learning content based on teaching objectives, and different cognitive levels reflect different question difficulties. On the other hand, due to the varying actual ability levels of learners, the perceived difficulty of the same question may differ. Therefore, when generating questions, it is important to consider not only the cognitive level involved in the question but also the difficulty of the question.

### Difficulty-controllable QGEd

The main principle of difficulty-controllable QGEd is to train the model on datasets with difficulty annotations for questions, using context information, answers, and difficulty as model inputs, and controlling the model to generate questions of a specified difficulty level by introducing relevant mechanisms. For example, *Shoaib et al. (2025)* manually annotated a large number of questions with difficulty levels and extensively collected questions that were already labeled with difficulty levels to construct a training dataset. It employed a conditional generative adversarial network (cGAN) model. By specifying conditions such

as difficulty levels and subject domains during the generation process, and leveraging a balanced training dataset that covered multiple difficulty levels, the model was able to learn the characteristics of questions at different difficulty levels. This enabled the generation of multiple-choice questions (MCQs) that met predefined difficulty requirements.

However, the cost of manually annotating a large number of questions with difficulty levels is substantial. As a result, more research has been inclined towards designing an automatic annotation scheme for question difficulty. *Gao et al. (2019)* utilized two distinct machine reading comprehension (MRC) models to simulate test-takers, labeling the difficulty of questions based on whether both MRC models could correctly answer the same question. This approach led to the construction of the first difficulty-annotated dataset based on the SQuAD dataset, and it enabled the implementation of difficulty-controllable QG. Although the difficulty annotation of questions in this study is straightforward, it significantly reduces costs and is relatively objective, providing a new perspective for constructing difficulty-annotated datasets.

For example, *Zhang, Xie & Qiu (2024)* trained various pre-trained language models (PLM) models on the CLOTH dataset to simulate test-takers of different proficiency levels, and the effectiveness of the difficulty control strategies is evaluated using IRT models. They presented a framework for generating difficulty-controllable multiple-choice cloze tests (MC cloze test) using PLM and item response theory (IRT). The framework estimates gap complexity *via* the Shannon entropy of candidate gap words and selects distractors of varying difficulty levels using PLM confidence score, semantic similarity, and Levenshtein distance. It also incorporates rules to reduce the generation of invalid distractors.

In other related research, multiple question-answering systems were trained to act as proxies for real learners. These systems were based on a variety of pre-trained models, such as BERT, RoBERTa, and DeBERTa, and were fine-tuned using data subsets of different magnitudes. The performance of these question-answering systems on the dataset was leveraged to determine the difficulty of the questions *via* item response theory (IRT). This calculated difficulty was a key factor in training a question-generation model that can produce questions with controllable difficulty (*Moon et al., 2024*; *Teruyoshi, Tomikawa & Masaki, 2024*; *Tomikawa & Masaki, 2024*; *Tomikawa, Suzuki & Uto, 2024*).

Some research has also attempted to incorporate knowledge tracing techniques, using the historical test results and reading comprehension question texts of specific learners to automatically predict and control the difficulty of generated questions (*Srivastava & Goodman, 2021*). These approaches must rely on specific subjects (automated models or real test-takers), and their results can vary greatly due to the different characteristics of the subjects, leading to limited scalability and hindering the exchange and comparison between different studies.

The absolute difficulty of a question is primarily measured by the characteristics of the question itself, without being influenced by the subject taking the test. In such studies, *Cheng et al. (2021)* defined the difficulty of a question as the number of reasoning steps required to answer it. They used graph-based methods to model logic, first generating an initial simple question based on the first node of the reasoning chain with a question generator (QGInitial), and then gradually increasing the reasoning steps through a

question rewriting module (QGRewrite) to transform the simple question into a more complex one. On this basis, they further increased the difficulty of the question by using two rewriting patterns: Bridge and Intersection. *Bi et al. (2024)* proposed a simple and intuitive cross-domain automatic difficulty evaluation technique that focuses solely on the complexity of the question itself, significantly reducing the impact of subjective factors from test-takers. This approach is conducive to the unification of question difficulty evaluation standards and greatly enhances the generalizability of absolute difficulty evaluation. Research on multiple-choice question generation, on the other hand, focuses on increasing the similarity of distractors to enhance the absolute difficulty of the questions (*Ch & Saha, 2023*; *Murakhovs'ka et al., 2021*).

### Cognitive level controllable QGEd

Cognitive level controllability emphasizes that questions should be able to cover different cognitive levels, from basic memory and understanding to higher levels of analysis, evaluation, and creation. When generating questions, it is important to consider the cognitive development stage of students and design questions that are appropriate for that stage. For example, for beginners, more foundational questions can be provided to consolidate knowledge; for advanced learners, more complex questions that require deep thinking can be designed to promote the development of their critical thinking and innovative abilities. There are two main schools of thought in academia regarding the classification of cognitive levels of questions. One is the delineation method proposed by *Graesser & Person (1994)*, and the other is the well-known Bloom's taxonomy of questions (*Bloom, 1956*). The former, based on extensive research of questions in psychology, education, and artificial intelligence, and according to the type of expected answers for educational purposes, identified sixteen common types. The latter classifies questions based on the cognitive processes required to obtain answers.

In both Graesser's and Bloom's classification methods, each question type is further divided into three common difficulty levels: low, medium, and high, as detailed in Table 4 (*Al Faraby, Adiwijaya & Romadhony, 2024*). For ease of understanding and discussion, these questions can further be generalized into factual and non-factual questions. Factual questions aim to retrieve specific factual information, such as names, locations, or time-related details. These types of questions are usually directly converted from specific sentences or information, and their answers can be found directly from the given context, involving relatively simple cognitive activities. Non-factual questions often require long and abstract answers, such as open-ended questions, detail reasoning questions, comparative analysis questions, *etc*. This category of questions tends to involve higher-order cognitive activities. The answers to non-factual questions may be in the given context or may not be in the given context.

1. *Factual QGEd*

Factual questions are the most commonly generated in QG research, and the technology is relatively mature. Early heuristic rule-based methods have been able to generate smooth, relevant, accurate, and controllable questions to a certain extent, with strong

**Table 4 The correspondence between cognitive level and difficulty levels.**

| Difficulty | Graesser | Bloom |
| --- | --- | --- |
| LOW | Verification | Recognition |
| | Disjunctive | Recall |
| | Concept completion | |
| | Example | |
| MIDDLE | Feature specification | Comprehension |
| | Quantification | |
| | Definition | |
| | Comparison | |
| HIGH | Interpretation | Application |
| | Causal antecedent | Analysis |
| | Casual consequence | Synthesis |
| | Goal orientation | Evaluation |
| | Instrumental/Procedural | |
| | Enablement | |
| | Expectation | |
| | Judgmental | |

interpretability. For example, *Wolfe (1976)* designed a pattern matching method for educational systems. *Lindberg et al. (2013)* incorporated semantic roles into template methods to generate questions supporting online learning. *Flor & Riordan (2018)* also generated factual questions based on semantic roles. However, these methods rely on domain expertise and experience, and their effectiveness depends on the variety and quantity of predefined rules or templates, making it difficult to handle more diverse and complex contexts that were not anticipated during the rule design process. Due to the inherent complexity of language, it is almost impossible to manually discover and summarize all question generation rules, which would require a huge manual effort. And rules designed for one domain are usually difficult to quickly transfer to other domains. Additionally, the construction of manual templates will also greatly limit the diversity of generated questions. Therefore, these methods are difficult to generalize and apply on a large scale.

With the advent of the deep learning era, neural network models quickly replaced heuristic rule-based methods. *Du & Cardie (2017)* were among the first to introduce attention mechanism-based sequential learning models into QG research, directly enabling models to generate questions based on the sentences containing the answers through Seq2Seq learning, demonstrating that NQG techniques outperform traditional heuristic rule-based approaches. However, the aforementioned appoach has two major issues: (1) It neglects the processing of specific answer information within sentences. (2) It neglects the processing of contextual information surrounding the sentences. These issues prevent the scheme from ensuring the relevance of generated questions to answers and context, leading to poor interpretability, and an inability to handle paragraph-level QG tasks.

To address issue (1), researchers have chosen to embed the answer location information into neural network models. For example, *Zhou et al. (2017)* encoded the BIO tags of the answer locations as real-valued vectors embedded into the encoder layer. *Zhao et al. (2018)* proposed a gated self-attention network encoder with answer marking and a maximum

output pointer to avoid generating repetitive questions. *Yang et al. (2017)* and *Yuan et al. (2017)* respectively added an additional binary feature to the word embeddings of paragraph markers and document markers. *Steuer, Filighera & Rensing (2020)* used answer-aware NQG technology, combining two new answer candidate selection strategies based on semantic graph matching, to generate clear, answerable, and educationally valuable factual questions. Although these approaches have made the relevance of generated questions to answers more intimate, they struggle to ensure the relevance of questions to context, and there is a problem of generating questions that contain the answer itself.

To address the aforementioned issues and issue (2), researchers have chosen to independently encode the answer and context and model their relationship, enabling the model to generate high-quality factual questions relevant to both answers and context. For instance, *Kim et al. (2019)* used a dual encoder to separately encode the answer and the paragraph text with answer masking, and during the question generation process, they used an Answer-Separated Decoder to combine context features from the paragraph (obtained through attention mechanisms) and key features from the target answer's keyword network, solving the problem of generating questions that contain answer information and improving the relevance of questions to paragraph context. *Wang et al. (2019)* designed a weakly supervised discriminator that encodes the answer and context separately to capture the relationship between the two. *Song et al. (2018)* used two independent bidirectional long short-term memories (Bi-LSTMs) to encode the context and answer, and then matched each hidden state in the paragraph with all hidden states of the answer through a multi-perspective context matching (MPCM) algorithm to gather context information related to the answer. *Zeng et al. (2022)* proposed a dual-attention-based paragraph-level QG model, which, although only encoding the overall paragraph, used attention mechanisms to capture the relationship information between the answer and the overall paragraph for both the paragraph and the sentence containing the answer.

### 2. Non-factual QGEd

The main approach in this type of research currently is to reconstruct or modify datasets to optimize input features for training specific models to generate non-factual questions. For example, inspired by Graesser's classification, *Cao & Wang (2021)* directly adopted the cognitive-based question classification scheme revised by *Olney, Graesser & Person (2012)* to construct a new dataset and successfully implemented technology for generating different cognitive levels of questions; *Jia et al. (2021)* reconstructed the original RACE dataset and achieved certain results in generating English examination questions; *Murakhovs'ka et al. (2021)* built a new QA dataset by mixing nine datasets with different answer types to train the MixQA model, successfully generating a diversified cognitive levels of questions; *Qu, Jia & Wu (2021)* compared and analyzed the RACE and SQuAD datasets in terms of question form structure, concluding that the RACE dataset is more suitable for QGEd, and built a model specifically for generating non-factual questions trained on the RACE dataset; *Stasaski et al. (2021)* used a causality extractor to construct a

causal question dataset to adjust neural networks, successfully generating causal questions from textbook texts.

There are also studies attempting to generate non-factual questions by modifying neural network models, such as (*Babakhani et al., 2024*) who fine-tuned the architectures of large language models flan-T5 and GPT-3, using Seq2Seq generation technology to automatically generate subjective questions for news media posts. *Qu, Jia & Wu (2021)* proposed a multi-agent communication model to iteratively generate and optimize questions and keywords, then use the generated questions and keywords to guide the generation of answers. However, the schemes for modifying neural network models to generate non-factual questions still do not have controllable specific cognitive levels. Therefore, datasets with cognitive level labels are of great significance for controllable QGEd research.

Some studies have leveraged the powerful content learning capabilities of LLMs, employing fine-tuning or prompt engineering to train or guide these models in generating high-quality questions. *Duong-Trung, Wang & Kravcik (2024)* fine-tuned ChatGPT-3.5-turbo using a training dataset comprising 1,026 master's course questions across 29 subjects, resulting in the development of BloomLLM. These data were annotated by experts according to the six cognitive levels of Bloom's Taxonomy, ensuring that the generated questions are semantically interrelated and aligned with educational objectives. Through this approach, BloomLLM is capable of generating questions that better meet educational needs, addressing the shortcomings of basic LLMs in semantic relevance and hallucination issues, thereby enabling more effective application of generative AI in the field of education. In terms of prompt engineering, *Lee et al. (2024)* developed a framework based on a 2D matrix, in conjunction with prompt engineering techniques for ChatGPT, to facilitate the automatic generation of English reading comprehension questions. Teachers can select appropriate question types and formats from the matrix according to the reading materials and learning objectives, and input corresponding prompts. ChatGPT will then generate the relevant questions. The generated questions cover a range of types, including literal comprehension and inferential judgment. *Tonga, Clement & Oudeyer (2024)* leveraged LLMs to simulate the roles of students and teachers. By designing specific prompts, it guided the teacher model to generate probing questions. The diversity and creativity of the generated prompts were controlled by adjusting the model's temperature parameter. These questions were heuristic in nature.

## Focus on teaching content

A question may belong to a higher cognitive level, but this does not automatically endow it with educational value. Questions with educational value should enable learners to gain knowledge, develop skills, and achieve teaching objectives or fulfill the purposes of teaching evaluation in the process of answering the questions. To generate questions with educational value, and to make the generated questions more educationally valuable, some studies focus on generating questions targeted at content with educational value within the context, and use richer information related to the answers as input to improve the quality

of question generation, such as the external knowledge background of the answers, semantic features, and positional features.

Regarding the introduction of external knowledge background information, *Liu & Zhang (2022)* utilized the Chinese Wikipedia knowledge graph (CN-DBpedia2) to extract knowledge triples related to the answers and used the pre-trained language model BERT to encode the target answers and the graph separately, enriching the answer feature information and enhancing the educational value of the generated questions. In terms of utilizing the semantic features of answers, *Zhou et al. (2017)* incorporated linguistic features such as part-of-speech, named entities, and word forms into the QG system's input; *Du & Cardie (2018)* improved the accuracy of questions by annotating each pronoun's antecedent and its coreference location features in the input text. Regarding the processing of answer location features, *Sun et al. (2018)* designed a position-aware model that enhances the model's perception of the context word positions by encoding the relative distance between context words and answers as position embeddings; *Cheng et al. (2021)* utilized open information technology to construct the original text into a directed graph composed of entity nodes and relationship edges, and then encoded it with graph neural networks, thereby leveraging the positional information of the answer nodes within the graph.

However, the above technologies, which focus on teaching content, all adopt answer-aware QG strategies, requiring the target answers to be manually annotated in advance. This undoubtedly increases the workload of human teachers in practical applications, reflecting the issue of insufficient technological intelligence level. Therefore, technologies that can generate answers and corresponding questions based on a given paragraph have been valued by researchers. This technology is known as joint question-answer generation (*Cui et al., 2021*). In joint question-answer generation, automatically selecting content with educational value from a given context is a key step. For example, *Chen, Yang & Gasevic (2019)* proposed nine strategies for selecting important sentences, including choosing sentences at the beginning of a paragraph, sentences containing new words, and sentences with the lowest similarity to other sentences. *Steuer et al. (2021)* used keyword filtering and classifiers to filter out sentences containing key concepts, and then used dependency parsing and semantic graph matching techniques to identify and extract content with complete semantic information and educational value from textbooks as target answers for generating questions. *Yao et al. (2022)* used heuristic rules to select key words that help understand the story, such as characters, emotions, and causal relationships. *Ch & Saha (2023)* identified key content in textbooks through word frequency statistics. In addition to rule-based and statistical methods, there are also studies that train models to automatically select sentences. For example, *Du & Cardie (2017)* used a hierarchical neural network framework to encode the input context at both the sentence level and paragraph level, treating sentences containing the target answer as the target sentences, and the given context as input, directly training a neural network model capable of automatically extracting sentences with question-asking value.

Recently, *Maity, Deroy & Sarkar (2025)* directly utilized education-related content as the data source, integrating retrieval-augmented generation (RAG) based on BART with prompt engineering for large language models. By combining the strengths of retrieval and few-shot learning, it generated higher-quality questions that better aligned with educational objectives.

## DATASETS

In this section, we will present an overview of the datasets frequently employed in QG and provide detailed introductions and summaries of several representative datasets.

The availability and characteristics of datasets are crucial for NQG systems. Based on the source of the dataset content, this study categorizes the surveyed public datasets into two major categories: non-education-related and education-related. Table 5 shows a comparison of these datasets in terms of the main cognitive levels of questions, question types, subjects (content sources), the presence of cognitive/difficulty classification labels, and the forms of given contexts. Specifically, in the cognitive level column, datasets labeled as containing non-factual questions indicate that the datasets include such questions, but this does not imply that all questions within the datasets are non-factual.

### Non-education related datasets

The content of non-education-related datasets is primarily sourced from Wikipedia, news articles, speech transcripts, or paragraph texts from the internet, rather than educational textbooks. Even so, some datasets that encompass non-factual questions and difficulty classification features also offer valuable resources for exploring how to generate questions that are of higher cognitive levels or differentiated in difficulty. For instance, HotpotQA (*Yang et al., 2018*) includes two modes: simple and difficult, with solving difficult mode questions requiring cross-document reasoning. NQ (*Kwiatkowski et al., 2019*) is divided into "simple, medium, difficult" difficulty levels based on the naturalness of the questions and the findability of the answers. ODSQA (*Lee et al., 2018*) may require reasoning and understanding based on colloquial questions. The questions in CoQA (*Reddy, Chen & Manning, 2019*) are generated based on a segment of situational dialogue stories, forming a logically coherent sequence of questions. This type of question is particularly beneficial for cultivating learners' language logical expression abilities. The questions in Cosmos QA (*Huang et al., 2019*) require common sense reading comprehension, mainly including questions about the possible causes or consequences of events.

### Education related datasets

Education related datasets not only focus on content that can be used for education as their data source but also more universally pay attention to the collection and processing of questions with different difficulties and more cognitive levels. Among them, ARC (*Clark et al., 2018*), although it includes a variety of question types, only simply divides the questions into an Easy set and a Challenge set. RACE (*Lai et al., 2017*), as described in section "THREE CORE OBJECTIVES," has achieved considerable results in various QGEd studies and is divided into two difficulty levels: the middle school RACE-M and the high

**Table 5 The datasets for QG including QGEd.**

| Datasets | | Cognitive levels | Question types | Subjects (content sources) | Cognitive/difficulty classification | Language |
|---|---|---|---|---|---|---|
| Non-education related | SQuAD (*Rajpurkar et al., 2016*) | Factual | Short answer | Wikipedia | × | English |
| | HotpotQA (*Yang et al., 2018*) | Non-factual | Short answer | Wikipedia | √ | English |
| | NewsQA (*Trischler et al., 2017*) | Factual | Short answer | Wikipedia | × | English |
| | MLQA (*Lewis et al., 2020*) | Factual | Short answer | Wikipedia | × | Multilingual |
| | XQuAD (*Artetxe, Ruder & Yogatama, 2020*) | Factual | Short answer | Wikipedia | × | Multilingual |
| | ODSQA (*Lee et al., 2018*) | Non-factual | Short answer | Speeches | × | English |
| | NQ (*Kwiatkowski et al., 2019*) | Non-factual | Short answer | Wikipedia | √ | English |
| | Cosmos QA (*Huang et al., 2019*) | Non-factual | Multiple choice | Common sense | × | English |
| | CMRC (*Cui et al., 2019*) | Factual | Short answer | Wikipedia | × | Chinese |
| | ELI5 (*Fan et al., 2019*) | Non-factual | Short answer | Web document | × | English |
| | CoQA (*Reddy, Chen & Manning, 2019*) | Non-factual | Short answer | Conversations | × | English |
| | DuReader (*He et al., 2018*) | Non-factual | Short answer | Web document | × | Chinese |
| Education related | OpenBookQA (*Mihaylov et al., 2018*) | Non-factual | Multiple choice | Science | × | English |
| | SciQ (*Welbl, Liu & Gardner, 2017*) | Factual | Multiple choice | Science | × | English |
| | MedMCQA (*Pal, Umapathi & Sankarasubbu, 2022*) | Non-factual | Multiple choice | Medicine | × | English |
| | RACE (*Lai et al., 2017*) | Non-factual | Multiple choice | Entrance exam | √ | English |
| | RACE_C (*Liang, Li & Yin, 2019*) | Non-factual | Multiple choice | English exam | √ | English |
| | EQG_RACE (*Jia et al., 2021*) | Non-factual | Multiple choice | English exam | √ | English |
| | EMBRACE (*Zyrianova, Kalpakchi & Boye, 2023*) | Non-factual | Multiple choice | English exam | √ | English |
| Education related | ReClor (*Yu et al., 2020*) | Non-factual | Multiple choice | English exam | × | English |
| | TabMCQ (*Jauhar, Turney & Hovy, 2016*) | Non-factual | Multiple choice | Multidiscipline | × | English |
| | EduQG (*Hadifar et al., 2023*) | Non-factual | Multiple choice | Multidiscipline | × | English |
| | CLOTH (*Xie et al., 2018*) | Non-factual | Fill-in-the-blank | Middle school test | √ | English |
| | INQUISITIVE (*Ko et al., 2020*) | Non-factual | Short answer | Multidiscipline | × | English |
| | ARC (*Clark et al., 2018*) | Non-factual | Multiple choice | Science | √ | English |
| | MAWPS (*Koncel-Kedziorski et al., 2016*) | Non-factual | Math word | Math | × | English |
| | Math23K (*Wang, Liu & Shi, 2017*) | Non-factual | Math word | Math | × | Chinese |
| | CSFQGD (*Zhang et al., 2021*) | Factual | Fill-in-the-blank | Chinese test | × | Chinese |
| | SocratiQ (*Ang, Gollapalli & Ng, 2023*) | Non-factual | Short answer | Multidiscipline | × | English |
| | LearningQ (*Chen et al., 2018*) | Non-factual | Short answer | Multidiscipline | √ | English |
| | FairytaleQA (*Xu et al., 2022*) | Non-factual | Short answer | Fairy tale | √ | English |

school RACE-H. CLOTH (*Xie et al., 2018*) categorizes questions into four types based on the cognitive effort required to answer them: Grammar, Short-term reasoning, Matching/ Paraphrasing, and Long-term reasoning, with the largest proportion of Long-term

reasoning questions. FairytaleQA divides questions into explicit and implicit categories based on whether the answers can be directly found in the text. In fact, the former belongs to factual questions, whereas implicit questions are non-factual questions that require summarization and judgment based on implicit information in the text.

Due to the large scale and high-quality real exam questions of RACE, it subsequently became one of the most commonly used datasets in QGEd, and some follow-up work has further supplemented RACE. In 2019, *Liang, Li & Yin (2019)* released the RACE-C, in which they collected real exam questions from Chinese university English reading comprehension, further enriching the difficulty range of RACE's questions. In 2021, based on RACE, *Jia et al. (2021)* released EQG-RACE specifically for educational QG tasks. EQG-RACE removed general questions not suitable for QG tasks and automatically annotated the key sentences of the questions by calculating the ROUGE-L between the answers and the article sentences. In 2023, *Zyrianova, Kalpakchi & Boye (2023)* conducted a detailed evaluation of RACE by scoring some data samples and manually annotated the key sentences of some questions, releasing EMBRACE.

Although these datasets have potential for QGEd research, the division of cognitive levels and difficulty in most datasets is empirical, lacking unified, clear, and quantifiable standards, which is not conducive to the development of controllable QGEd with cognitive levels and difficulty. To achieve more fine-grained controllable QGEd, a more specific and scientific classification and annotation of the cognitive levels and difficulty of questions are still needed. For example, LearningQ (*Chen et al., 2018*) uses the Bloom's Taxonomy introduced in section "THREE CORE OBJECTIVES" to annotate and categorize questions, but only 200 random questions in the dataset have this classification label, which is far from meeting the training needs of NQG.

In fact, according to section "THREE CORE OBJECTIVES," recent outstanding research in the field of QGEd has conducted extensive work on self-built datasets. These efforts essentially involve either automatic or manual annotation of training data with difficulty levels or cognitive hierarchies. This highlights the significant role of question difficulty and cognitive hierarchy, as two educational attributes, in the construction of datasets specifically dedicated to QGEd research.

## EVALUATION SCHEMES

In this section, we focus on presenting the main evaluation schemes and criteria for QGEd systems. Our article primarily introduces these criteria rather than analyzing the State-of-the-Art (SoTA) outcomes, which are the results obtained through these criteria. We believe that establishing clear evaluation standards is crucial as it provides a unified framework for measuring progress and innovation. This allows researchers to better understand and compare different approaches within the field. There are two commonly used evaluation schemes in QGEd from two aspects: automatic schemes and manual schemes.

Question quality evaluation is a critical component in determining the success or failure of QGEd research. This section provides a brief overview of the current automatic and manual schemes used for evaluating QGEd systems.

## Automatic schemes

Currently, most QGEd studies employ automatic evaluation metrics inherited from traditional QG research, with BLEU (*Papineni et al., 2002*), ROUGE (*Lin, 2004*), METEOR (*Banerjee & Lavie, 2005*), chrF (*Popović, 2015*) and BERTScore (*Zhang et al., 2020*) being the primary ones.

1. *BLEU*

BLEU is an n-gram precision-based evaluation metric that evaluates quality of generated text by comparing the n-gram overlap between generated text and a set of reference texts, with the specific calculation method as shown:

$$\text{BLEU} = \text{BP} \cdot \exp\left(\sum_{n=1}^{n} w_n \, \log \, p_n\right). \tag{2}$$

Here, $p_n$ represents the precision of the nth gram, $w_n$ is the corresponding weight, which is typically a uniform weight in BLEU, meaning that the weight of each n-gram is the same. BP stands for the brevity penalty factor, which comes into effect when the length of the generated text is less than or equal to the length of the reference text.

2. *ROUGE*

ROUGE evaluates the quality of generated text by comparing the overlapping units (such as n-grams, word sequences, and word pairs) between machine-generated text and ideal reference texts. It includes a total of four metrics: ROUGE-N, ROUGE-L, ROUGE-W, and ROUGE-S. The value of ROUGE-N is obtained by dividing the number of shared n-grams by the total number of n-grams in the reference summary. This ratio reflects the similarity in content between the generated text and the reference text. The specific calculation method is shown as:

$$\text{ROUGE} - \text{N} = \frac{\sum_{S \in \{RefrenceSummaries\}} \sum_{gram_n} Count_{match}(gram_n)}{\sum_{S \in \{RefrenceSummaries\}} \sum_{gram_n} Count(gram_n)}. \tag{3}$$

This formula calculates the n-gram recall rate, which is the ratio of the number of common n-grams between the machine-generated text and the reference text to the total number of n-grams in the reference text.

ROUGE-L is an evaluation method based on the longest common subsequence (LCS). It seeks the longest sequence of identical words between two sequences (generated text and reference text). The longer the LCS, the more similar the two texts are. ROUGE-L uses the F-measure (a metric that combines precision and recall) to evaluate similarity. ROUGE-W is an evaluation method for weighted longest common subsequence (weighted LCS). It is similar to ROUGE-L but adds weighting for consecutive matching words, meaning that if two words are consecutively matched, they will receive a higher score. ROUGE-S is based on the statistics of Skip-Bigrams. Skip-Bigrams are pairs of words that appear in a sentence but can have any number of other words in between. In summary, ROUGE focuses more on the recall of correct information in the generated text compared to BLEU when

evaluating the similarity between generated text and reference text, that is, how much information from the reference text is included in the generated text.

3. *METEOR*

METEOR is an evaluation metric that takes into account synonyms, morphological variations, and sentence structure. Its calculation is more complex, involving the alignment of words in the generated text and the reference text, and calculating a score based on matches. The calculation formula is shown as:

$$Fmean = \frac{10PR}{R+9P}$$
$$Penalty = 0.5 * \left(\frac{\#chunks}{\#unigrams_{matched}}\right)^3 \tag{4}$$
$$\text{METEOR Score} = Fmean * (1 - Penalty).$$

METEOR can calculate the similarity between generated questions based on word-level precision and recall, as well as a penalty for word order. The *Fmean* is the harmonic mean of unigram precision *P* and unigram recall *R*, with a higher weight given to recall as *9R*. The *Penalty* calculates the degree of fragmentation of matches due to word order disruption; if words in the generated text match with words in the reference text but are out of order, the *Penalty* increases.

4. *ChrF*

ChrF (Character n-gram F-score) is an evaluation metric based on character-level assessments, primarily used to evaluate the quality of text generation tasks. It assesses the similarity between reference and generated texts by calculating the matching counts of character-level n-grams. The core advantages of chrF include: (1) It can better capture syntactic and semantic information in morphologically rich languages; (2) It is language-independent, as it does not rely on language-specific tokenization processes and is applicable to multiple languages; (3) It is both simple and efficient, requiring no additional tools or knowledge sources. The general formula for the chrF score is:

$$\text{chrF}\beta = \left(1 + \beta^2\right) \frac{\text{chrP} \cdot \text{chrR}}{\beta^2 \cdot \text{chrP} + \text{chrR}}. \tag{5}$$

Here, chrP and chrR represent the precision and recall of character n-grams, calculated as the average across all character n-grams. The parameter $\beta$ is used to adjust the relative importance of recall compared to precision, specifically giving $\beta$ times more weight to recall. This metric is language-independent and token-agnostic, making it highly versatile and widely applicable across various translation tasks.

5. *BERTScore*

BERTScore is a text similarity evaluation metric based on the BERT pre-trained model, designed to measure the semantic similarity between generated text and reference text. It leverages contextual embeddings, utilizing the context-aware word vectors generated by the BERT model to capture semantic information. This approach enables BERTScore to

demonstrate strong semantic matching capabilities, outperforming traditional evaluation methods based on n-gram overlap (such as BLEU) in capturing semantic similarity. Additionally, BERTScore supports evaluation across multiple languages, further enhancing its applicability in diverse linguistic contexts.

The calculation process of BERTScore is as follows: (1) Encode the candidate and reference texts using the BERT model to obtain contextual embeddings for each word; (2) Calculate the cosine similarity between each word in the candidate text and each word in the reference text; (3) Use a greedy matching algorithm to find the most similar word in the reference text for each word in the candidate text; (4) calculate precision, recall, and F1-score based on the matching results. Precision quantifies the proportion of words in the generated text that match the reference text, while recall measures the proportion of words in the reference text that match the generated text. The F1-score serves as a balanced metric, taking into account both precision and recall.

6. *Other metrics*

The aforementioned several metrics are commonly used in traditional QG research to evaluate the natural fluency of generated questions, focusing on the relevance and answerability of the generated questions, but they cannot directly evaluate the cognitive levels, difficulty, and diversification of the questions, which are the quality dimensions of more concern in the field of education. Researchers, considering this, have begun to explore other metrics for automatically evaluating the quality of generated questions. In terms of evaluating the cognitive level of questions, *Stasaski et al. (2021)* designed the Cause/Effect Presence metric to specifically evaluate the quality of causal questions. When the expected answer to a generated question is a "cause," the Cause Presence metric checks whether the "effect" extracted from the text is mentioned in the question. For questions where the "cause" is the expected answer, the size of the intersection between the words of the extracted "effect" appearing in the question and the set of "cause" words is calculated, then divided by the size of the "cause" word set to obtain the cause recall, and the same goes for Effect Presence. *Steuer, Filighera & Rensing (2020)* used the frequency of Wh-words to make a rough evaluation of the cognitive level of questions.

For automatic difficulty evaluation, please refer to section "Difficulty-controllable QGEd," which can be divided into absolute difficulty evaluation (*Gao et al., 2019*; *Srivastava & Goodman, 2021*) that only considers the complexity of the question itself, and relative difficulty evaluation (*Bi et al., 2024*; *Cheng et al., 2021*) that is calculated in combination with the test-taker's ability.

It is mainly evaluated by inversely evaluating similarity to evaluate the diversification of the questions, that is, the degree of difference in expression. For example, if a generated question is not similar to any question in the reference set, it can be considered novel. *Srivastava & Goodman (2021)* used novelty to evaluate question quality, which essentially uses text similarity metrics (such as cosine similarity, Jaccard similarity, *etc.*) to evaluate the difference between each generated question and the questions in the reference set. *Wang et al. (2020a)* and *Gou et al. (2023)* both used the pairwise metric to calculate the

average BLEU-4 score between any two questions in the generated question set. A lower pairwise indicates a higher degree of question diversity.

In summary, automatic schemes focus on evaluating the similarity between the generated questions and the target questions, while relying on manual evaluation for quality dimensions more relevant to education, such as difficulty, cognitive level, and diversity. This inevitably introduces subjectivity from the evaluator, affecting the objectivity of the evaluation, which is not conducive to comparison and exchange among similar studies, and also increases research costs and time, affecting research efficiency. Although some studies have designed other automatic evaluation schemes for the cognitive level and diversity of questions, these schemes are still based on semantic similarity evaluation, whereas factors such as the complexity of knowledge, the degree of association between question answers and context have a greater impact on cognitive levels than semantic similarity between questions.

### Manual schemes

Automatic evaluation holds significant advantages in terms of research efficiency and cost. However, existing reliable evaluation strategies are largely confined to the realm of semantic similarity assessment. When it comes to evaluating the quality of questions for educational applications, greater emphasis should be placed on dimensions such as educational value embedded in the questions. In this regard, human evaluation by educational experts still plays an irreplaceable role.

Manual evaluation typically covers three main criteria: naturalness (fluency, relevance, and answerability), difficulty, and helpfulness (educational value), which may be expressed differently or include other standards in different studies. The evaluation method usually involves experts, crowd-sourced participants, or the authors themselves, who annotate the generated questions based on specific metrics, and finally, the evaluation results are derived through statistics. For example, in the research (*Gao et al., 2019*), the difficulty of questions is divided into three levels: 1 (easy), 2 (medium), and 3 (difficult), requiring evaluators to score each question. In the research (*Wang et al., 2018*), evaluators are asked to mark each question as "true/false" for fluency (coherent and grammatically correct), question-answer relevance, and machine generation. *Steuer et al. (2021)* invited three trained education experts to evaluate question quality based on the annotation scheme developed by *Horbach et al. (2020)*, which consists of 6 binary indicators, 2 ternary indicators, and 1 quinary indicator, covering various aspects such as understandability, content relevance, naturalness, educational value, and cognitive levels of the questions. Obviously, manual evaluation is time-consuming and labor-intensive, requires high professional knowledge and experience from evaluators, and due to the subjectivity of the evaluation method, it is difficult to unify standards, leading to inconsistent evaluation results and limited widespread application.

## DISCUSSIONS

As a branch of QG in the field of education, QGEd has already established a considerable research foundation and progress. With the continuous advancement of technology,

especially breakthroughs in the fields of artificial intelligence and natural language processing, QGEd is gradually revealing its immense potential in educational innovation. For instance, the integration of deep learning models like transformers has enabled the generation of more complex and contextually relevant questions, moving beyond the limitations of traditional rule-based systems (*Vaswani et al., 2017*).

However, to meet the demands of the educational field, QGEd also faces numerous challenges. Among these, multimodal data processing stands out as a critical area. Educational questions should not be limited to text but should incorporate various forms such as images, audio, and video. While prior studies have explored visual question generation (VQG) (*Mulla & Gharpure, 2023*), these approaches often focus on general images rather than structured educational diagrams like mathematical geometric figures or flowcharts. Future research could explore how to adapt VQG techniques to better suit the specific requirements of educational content.

The controllability of fine-grained cognitive level and difficulty is another significant challenge. Questions should be designed to target specific cognitive abilities and difficulty levels to effectively facilitate learning. Although item response theory (IRT) has been widely used for question difficulty estimation (*Bi et al., 2024*), it often fails to capture the nuanced cognitive processes involved in answering questions. Recent studies have attempted to align question generation with established taxonomies like Bloom's Taxonomy (*Krathwohl, 2002*). For example, *Duong-Trung, Wang & Kravcik (2024)* proposed BloomLLM, which combines supervised fine-tuning with Bloom's Taxonomy to generate questions targeting specific cognitive levels.

The construction of specialized educational datasets is also crucial. Most existing datasets are either too generic or lack the necessary metadata for educational applications (*Al Faraby, Adiwijaya & Romadhony, 2024*). This limitation affects the quality and relevance of generated questions. Future work should focus on creating datasets that not only include questions and answers but also provide detailed annotations regarding educational objectives, cognitive levels, and difficulty. This would require collaboration between technologists and educational experts to ensure that the datasets meet the specific needs of educational settings.

Automatic evaluation technologies must be further developed to accurately assess the quality of generated questions. Traditional metrics like BLEU and ROUGE (*Papineni et al., 2002*; *Lin, 2004*) are limited in their ability to evaluate the educational value and cognitive level of questions. While newer metrics like BERTScore (*Zhang et al., 2020*) offer improvements by considering semantic similarity, they still fall short in capturing the educational significance of questions. Future research should focus on developing evaluation frameworks that incorporate educational theories and cognitive science principles.

System architecture design for QGEd systems also requires special attention. Unlike traditional QG systems, QGEd systems need to handle complex educational content and generate questions that align with specific teaching objectives. *Al Faraby, Adiwijaya & Romadhony (2024)* proposed a system architecture (as shown in Fig. 4) that includes key content identification, answer identification, and question construction. However, this

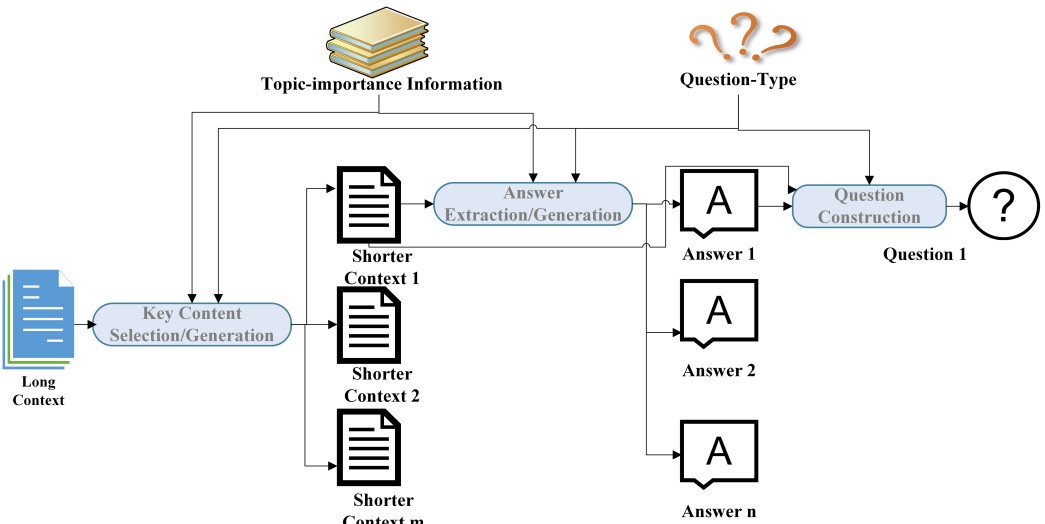

**Figure 4** The structure diagram of the QGEd system for generating text questions based on text data.

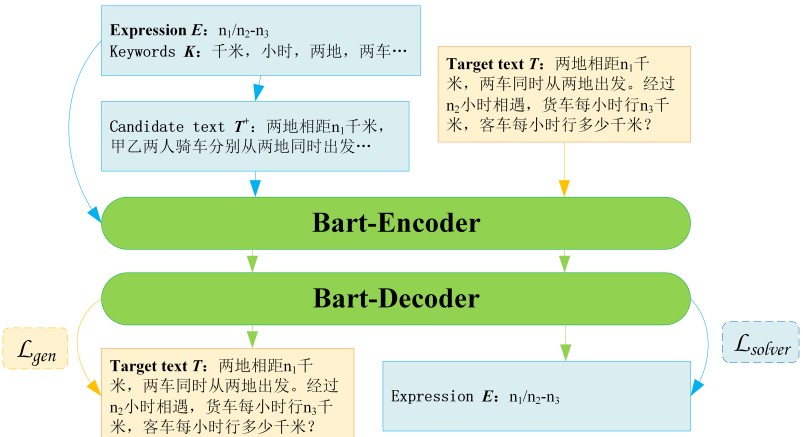

**Figure 5** The model framework for generating math word questions.

architecture primarily focuses on text-based questions. For STEM education, a more specialized architecture is needed that can handle complex problems involving multiple knowledge points and real-world scenarios. *SHEN (2022)* proposed a mathematical problem generation framework (as shown in Fig. 5. The diagram depicts a junior high math application problem where two cars travel towards each other and meet.千米: kilometer, 小时: hour, 两地: two places, 两车: two cars) that combines mathematical formulas with textual descriptions. This approach could serve as a foundation for developing more sophisticated QGEd systems for STEM subjects.

Looking ahead, with the continuous iteration and progress of general-purpose large language model technologies, cost-effective and efficient techniques such as fine-tuning of

large models, prompt engineering, and retrieval-augmented generation (RAG) are likely to become hotspots in QGEd research. Meanwhile, the development of specialized datasets for QGEd should also be given high priority. Technologists should deepen their collaboration with educational researchers to further integrate QGEd technologies with educational theories, thereby ensuring that the generated questions can truly and accurately serve students' learning and cognitive development.

## CONCLUSION

This review introduces the relationship between QGEd and QG, defines the three core objectives of QGEd—diversity, controllability, and focus on teaching content. We then analyze the technological evolution of QG, dividing it into heuristic rule-based and neural network technology stages, and discuss QGEd's evolution from 2015 to 2024. Based on this technological progression, we overview QGEd's current state across its three core objectives, datasets, and evaluation methods. We also identify pivotal challenges like the need for multimodal data processing, controllability of cognitive and difficulty levels, and specialized dataset development. To address these, we suggest future research directions, emphasizing automatic evaluation technology and system architecture design. Overall, this review offers valuable insights for researchers and practitioners, summarizing existing research and providing a foundation for QGEd's advancement.

### Funding

The work is funded by the Science Foundation of Ministry of Education of China (No. 24YJA880007) and the National Natural Science Foundation of China (No. 62167007). The funders had no role in study design, data collection and analysis, decision to publish, or preparation of the manuscript.

### Grant Disclosures

The following grant information was disclosed by the authors:
Science Foundation of Ministry of Education of China: 24YJA880007.
National Natural Science Foundation of China: 62167007.

### Competing Interests

The authors declare that they have no competing interests.

### Author Contributions

- Xiaohui Dong conceived and designed the experiments, performed the computation work, authored or reviewed drafts of the article, and approved the final draft.
- Xinyu Zhang conceived and designed the experiments, performed the experiments, performed the computation work, prepared figures and/or tables, authored or reviewed drafts of the article, and approved the final draft.
- Zhengluo Li performed the experiments, analyzed the data, prepared figures and/or tables, and approved the final draft.

- Quanxin Hou analyzed the data, authored or reviewed drafts of the article, and approved the final draft.
- Jixiang Xue analyzed the data, authored or reviewed drafts of the article, and approved the final draft.
- Xiaoyi Li analyzed the data, authored or reviewed drafts of the article, and approved the final draft.

## Data Availability

This is a literature review.

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
