# Peer review of "A literature review of research on question generation in education"

_PeerJ Computer Science, doi:10.7717/peerj-cs.3203_

## Round 0.1 · original submission · Major Revisions

Both reviewers have made relevant comments, please respond to them in detail

·

Basic reporting

- The ‘abstract’ could be made clearer. The transition between different sections could be smoother. Instead of starting a new sentence with "First," "Then," and "Finally," consider using more seamless transitions. Emphasize the contribution of the paper more clearly. Instead of just presenting the state of research, make it clear how this paper adds to the existing body of knowledge.

- Occasional grammatical errors and repetitive phrases should be addressed.

- The ‘Introduction’ section lacks adequate citations to support key claims and contextual statements. Adding references to foundational and recent works in Question Generation in Education (QGEd) would strengthen the scholarly grounding of the introduction.

- One issue that needs to be addressed is the inconsistent capitalization throughout the manuscript. For example, in lines 51 - 54 "In answer-aware QG, questions are formulated towards a given answer within the context, as shown in Table 1, where the Input Paragraph represents the provided context, and the Answer Span denotes the positional characteristics of the target answer within the given context" is not capitalized correctly. The terms "Input Paragraph" and "Answer Span" should be consistently capitalized if they are defined as specific components in the methodology. Ensure that all technical terms, sections, and model-specific terms are consistently capitalized or not, depending on their use.

- To improve readability and ensure smoother transitions between sections, it is recommended to include a brief introductory line at the beginning of each section or subsection. This line should provide an overview of what the section or subsection will cover. For example, instead of diving directly into the content, starting with a line like "This section describes the methodology used in..." or "In this section, we concentrate..." will guide the reader and enhance the flow of the manuscript. These introductory lines will also help the reader anticipate the content and structure of each section.

- Pay attention to sentence structure, punctuation, and overall grammar. There are a few instances where sentences are overly complex, and breaking them into simpler statements would enhance readability.

- In the 'Automatic Schemes' subsection, points ‘(3)’ (line 629) and ‘(4)’ (line 651) are repeated, indicating a need for better organization and proofreading.

Experimental design

- The manuscript effectively covers a broad range of literature, mapping the evolution of research in QGEd. However, it lacks a sufficiently critical analysis of seminal works in sections such as ‘Development of Technology’ and ‘Datasets’. More in-depth examination of foundational studies is essential. This should include a discussion of each work’s methodologies, key contributions, and limitations. Highlighting unresolved issues in prior research and explaining how this review addresses these gaps would strengthen its scholarly contribution and provide clearer insights into the field's development.

- It is important to first discuss ‘question types’ (with proper citations) before addressing the ‘development of technology’. This would provide a clearer context for understanding the evolution of QGEd research. Additionally, recent works in the development of QG technologies should include types of questions generated by each work cited. This will further clarify the specific contributions of each study and illustrate the focus of the research.


- While the manuscript covers a wide range of relevant works, foundational studies on pre-trained language models and large-scale dataset evaluations seem underrepresented. Including these would provide a more comprehensive perspective. Additionally, there are several recent works in the automated question generation (AQG) educational domain that explore state-of-the-art large language models (LLMs). The manuscript would benefit from a detailed discussion of these works, as their inclusion would provide a more up-to-date and thorough overview of the advancements in this area.

- There are a few automated evaluation metrics such as ChRF, BERTScore, etc., that are missing. It would be important to include these metrics in the ‘Automatic schemes’ section, as they are commonly used in question generation evaluation. Furthermore, it is essential to discuss the limitations of automated evaluation metrics and explain why manual evaluation is necessary in the ‘Evaluation Schemes’ section.

- It is advised to specify whether the works incorporated Bloom's Revised Taxonomy (or any other taxonomy). Doing so will help clarify the cognitive levels addressed by each study and provide a more comprehensive view of how QGEd research aligns with educational frameworks.

Validity of the findings

The ‘Conclusion’ section outlines key challenges comprehensively but could benefit from a more concise and structured presentation.

e.g.,

(i) Focus on summarizing the cognitive demand of question generation and the need for fine-grained controllability in a more concise way. Mentioning specific studies like Graesser & Person (1994) and Bloom (1956) is more suitable for the main text, not the conclusion.

(ii) Focus on summarizing the relevance of SHEN (2022)’s approach briefly, emphasizing how it contributes to the broader QGEd field. End with a forward-looking statement about potential advancements inspired by such frameworks.

Additional comments

It is important to add gloss in English to the Chinese text in Fig. 5, as all readers may not be able to read or understand Chinese. Providing an English translation or explanation would make the figure more accessible to a wider audience.

Reviewer 2 ·

Basic reporting

No comment

Experimental design

The survey design and methodology is described clearly: the selection of the databases, the search strategy etc. This is useful to be able to replicate the paper and to understand the followed process.

The article is a review which is focused on Question Generation in the education domain. Although it mentions rule- and template-based methods, the related work is primarily focused on the use of neural networks which are the current SoTA. I get this point, and thus, I would expect some related work on prompting techniques such as zero-shot and chain-of-thought that are widely use in other related NLP tasks and are starting to be used in generating high-quality questions, including open-ended and multiple choices.

It could be interesting to not only mention the related works but also to give a summary of current SoTA regarding QG. After reading the article, it is clear which are the different approaches but you don’t get how successful they are generating questions in terms of quality, diversity etc. Which are numbers that current systems obtains in the datasets mentioned in the survey? Using which evaluation metrics? Maybe you could try to enrich table 3 or create a new table or section summarising this information.

The paper is mainly focused in systems that generate English questions and mentioned a few for Chinese. There are systems that generate questions to languages other than English and Chinese. To me, the paper should clearly state which are the target languages of the survey and why.

A section including the limitations of the existing methods and future directions is not critical, but would give more value to the work. For instance, ideas such as the plausibility of generating biased questions, the use of multimodal questions, or including humans in-the-loop would show the future of such systems.

Validity of the findings

No comment

Additional comments

The paper is well written and easy to follow. However, there are a few mentions to computer vision at the beginning of the paper that does not fit with the scope of the paper. Authors should double check this.

Cite this review as

---

## Round 0.2 · Minor Revisions

Thanks for your submission. Kindly provide the revised version. One suggestion from my side, please include the contribution of the paper and application (this could be a section or before the conclusion. Feel free to add it where it adds most value).

**Language Note:** The review process has identified that the English language must be improved. PeerJ can provide language editing services - please contact us at [email protected] for pricing (be sure to provide your manuscript number and title). Alternatively, you should make your own arrangements to improve the language quality and provide details in your response letter. – PeerJ Staff

·

Basic reporting

* The manuscript demonstrates significant improvements in clarity, structure, and grammatical accuracy. The addition of citations and a clearer flow between sections enhances its scholarly quality.

* However, some grammatical issues persist, and inconsistencies in the capitalization of technical terms require attention.

* The authors have added valuable references to foundational works, but could further strengthen the context by discussing more recent advancements in large language models (LLMs).

* Although the abstract is concise, its transitions remain abrupt, and the manuscript’s unique contribution to question generation in education (QGEd) could be articulated more explicitly.

Experimental design

* The inclusion of ChrF and BERTScore addresses evaluation concerns, and discussions on the limitations of automated metrics are a welcome addition.

* The alignment with Bloom’s Revised Taxonomy is well-structured, yet the manuscript could benefit from a deeper critical analysis of foundational studies and methodologies.

* Expansion on how current advancements in LLMs enhance QGEd would further improve the study’s comprehensiveness.

Validity of the findings

The conclusion section has been revised for brevity and effectively summarizes key findings and challenges in QGEd. References to educational frameworks and previous studies have been integrated into the main text, addressing earlier concerns.

Additional comments

The manuscript has addressed most review comments adequately, significantly improving its readability, scholarly grounding, and overall structure. However, to ensure readiness for publication, the authors should:

* Address residual grammatical issues and ensure consistent capitalization of technical terms throughout the manuscript.


* Provide a more critical analysis of foundational studies and further expand the discussion of recent LLM advancements.

Reviewer 2 ·

Basic reporting

No comment

Experimental design

My main comments are covered (or answered), and it is a clearer paper now from my point of view.

Some of my comments have not been included, but the authors answered them and made clear why they don't include them. However, I would:

a) Explicitly say that the review is mainly focused on English QG.
b) Again, consider that the paper is mainly focused on presenting the main evaluation criteria rather than explaining which are the SoTA results nowadays. If this aspect is not part of the paper, I would also mention this explicitly (and why)

Validity of the findings

No comment

Additional comments

A final reading of the paper would be beneficial.

As a minor thing, I still don't understand why the intended audience is from computer vision.

Cite this review as

---

## Round 0.3 · Minor Revisions

Thanks for the submission of the revision. My feedback is addressed but not completely. You need to include a section about the contributions of your paper, both theoretical and practical. The conclusion needs to be revised as here you mention key take away points based on your results. In discussion, you need to explain your results with respect to existing literature, as discussion is to "move from results to meaningful discussion".

**PeerJ Staff Note**: Please ensure that all review, editorial, and staff comments are addressed in a response letter and that any edits or clarifications mentioned in the letter are also inserted into the revised manuscript where appropriate.

---

## Round 0.4 · accepted · Accept

Thanks a lot for the revision. I am happy to accept it. In final files, do minor edit. The contribution section should be written in third person. In other words, remove "we" and replace it with "The research...".

Reviewer 2 ·

Basic reporting

No comment

Experimental design

No comment

Validity of the findings

No comment

Additional comments

No comment

Cite this review as